# Unlearn What You Want to Forget: Efficient Unlearning for LLMs

**Jiaao Chen**
Georgia Institute of Technology
`jiaaochen@gatech.edu`

**Diyi Yang**
Stanford University
`diyiy@cs.stanford.edu`

## Abstract

Large language models (LLMs) have achieved significant progress from pre-training on and memorizing a wide range of textual data, however, this process might suffer from privacy issues and violations of data protection regulations. As a result, the ability to easily remove data related to individual users from such models while not deteriorating their predictive quality after the removal becomes increasingly important. To address these issues, in this work, we propose an efficient unlearning framework that could efficiently update LLMs without having to retrain the whole model after data removals, by introducing lightweight unlearning layers learned with a selective teacher-student objective into the transformers. In addition, we introduce a fusion mechanism to effectively combine different unlearning layers that learns to forget different sets of data to handle a sequence of forgetting operations. Experiments on classification and generation tasks demonstrate the effectiveness of our proposed methods compared to the state-of-the-art baselines[1].

## 1 Introduction

Utilizing Large Language Models (LLMs) has become the dominant paradigm for various NLP applications (Brown et al., 2020; Chowdhery et al., 2022a; Kojima et al., 2022; Ouyang et al., 2022; Brown et al., 2020; Radford et al., 2019; Lewkowycz et al., 2022; Qin et al., 2023; Touvron et al., 2023) as LLMs memorize a vast amount of knowledge during pre-training or fine-tuning on a wide range of textual data (Brown et al., 2020; Radford et al., 2019; Hoffmann et al., 2022; Webson and Pavlick, 2022; Min et al., 2022; Liang et al., 2022; Carlini et al., 2022). However, these data could contain sensitive information such as names, phone numbers, email addresses, and private clinical notes (Jang et al., 2022; Kurmanji et al., 2023;

Kumar et al., 2022).Extensive studies showed that LLMs could generate private information such as the Editor-in-Chief of MIT Technology Review including his family members, work address, and phone number (Carlini et al., 2022). Recently, the EU's General Data Protection Regulation (GDPR) and US's California Consumer Privacy Act (CCPA) have also required the *right to be forgotten*, introducing new regulations that require applications to support the deletion of user-generated content when requested by users (Sekhari et al., 2021; Kumar et al., 2022). In light of this, it is essential to provide LLMs with an efficient and effective way to unlearn the information requested by users.

Recent attention has been paid to the handling of such unlearning requests for LLMs through retraining and data pre-processing like SISA (Bourtoule et al., 2021; Kumar et al., 2022) where training data is stored in different isolated slices and each checkpoint is saved after training on each slice. When a deletion request is received, the respective data point will be removed from the slice, and the model checkpoint up to the data point will be used to further retrain the model. The effect of unlearning is often reflected by the model errors on the deleted data (models cannot predict the deleted data) (Kurmanji et al., 2023; Jang et al., 2022). Other works have also explored the design of algorithms that ensure differential privacy (DP) (Yu et al., 2021; Li et al., 2021; Anil et al., 2021). However, machine unlearning approaches like SISA (Bourtoule et al., 2021) usually require a significantly large amount of storage space (Bourtoule et al., 2021), and DP methods could result in a slow convergence and significant deterioration in model performance (Nguyen et al., 2022). In addition, both of them require retraining the whole model, which is extremely expensive and time-consuming considering the model scales of the current LLMs. These limitations also make them unable to dynamically deal with a sequence of unlearning requests which

---

[1]The codes are avaiable here: `https://github.com/SALT-NLP/Efficient_Unlearning/`

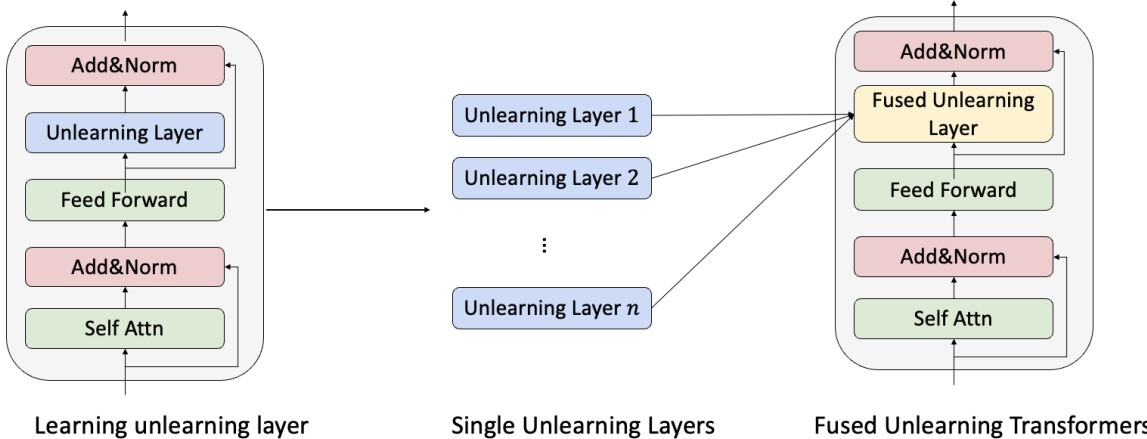

Figure 1: Overall process of our EUL framework. The unlearning layers are plugged into transformer layers after the feed-forward networks. During training, only the unlearning layers are learned to forget requested data while the original LLMs remain unchanged. For every deletion request, an unlearning layer is learned first and then merged with other unlearning layers via our designed fusion mechanism to form the fused unlearning transformer which satisfies a series of deletion requests.

is often the need in real-world scenarios (Jang et al., 2022; Nguyen et al., 2022).

To fill in these gaps, in this work, we propose an **E**fficient **U**nlearning method for **LLM**s (EUL) to efficiently unlearn what needs to be forgotten without completely retraining the whole model while retaining the performances of the models. Specifically, we propose a lightweight approach to learning the unlearning layer that is plugged into transformers through a selective teacher-student formulation (Kurmanji et al., 2023) within several updates, without tuning the large language models. Additionally, we introduce a fusion mechanism to effectively combine the weights of different unlearning layers that learn to forget different sets of data to a single unified unlearning layer by minimizing a regression objective. This allows EUL to efficiently address a sequence of deletion operations. To demonstrate the effectiveness of our proposed EUL, we perform experiments on IMDB (Maas et al., 2011) and SAMSum (Gliwa et al., 2019) in different settings compared to the state-of-the-art unlearning or model editing baselines. To summarize, our main contributions are threefold:

- We introduce an efficient unlearning method to remove the effect of required data in a lightweight way via a selective teacher-student formulation.

- We design a fusion mechanism to merge unlearning layers that are learned to forget different sets of data into a single unlearning layer to deal with a sequence of removal operations.

- We conduct experiments on classification and generation tasks with backbone models of different scales in different settings, to illustrate the effectiveness of EUL.

## 2 Related Work

### 2.1 Large Language Models

Large language models have witnessed extensive progress recently (Brown et al., 2020; Radford et al., 2019; Smith et al., 2022; Rae et al., 2021; Chowdhery et al., 2022b; Touvron et al., 2023), especially in terms of scaling up LLMs such as LLAMA (Touvron et al., 2023), Megatron-turing NLG (Smith et al., 2022), Gopher (Rae et al., 2021), and PaLM Chowdhery et al. (2022b). Other works have also achieved better performance with smaller models through longer training (Hoffmann et al., 2022), instruction tuning (Wang et al., 2022; Zhou et al., 2023) and human feedback (Ouyang et al., 2022). However, recent studies have shown that training data, such as personally identifiable information like names, phone numbers, email addresses, and even bank account numbers (Carlini et al., 2021; Lee et al., 2021; Carlini et al., 2022; Jagielski et al., 2022), can be easily extracted from LLMs because LLMs memorize the training data in billions of parameters (Carlini et al., 2022). Our work is proposed to alleviate such issues by allowing efficient unlearning of the requested or private data from the learned parameters in LLMs.

## 2.2 Machine Unlearning for Privacy

To mitigate the privacy risks for LLMs, machine unlearning methods have been introduced to remove the contributions of training examples that users request to be erased by users (Bourtoule et al., 2021; Chien et al., 2023) including exact unlearning that retrains deep learning models on new datasets after removal (Bourtoule et al., 2021) and approximate unlearning (Izzo et al., 2021; Golatkar et al., 2020; Kurmanji et al., 2023; Jang et al., 2022) which aims to modify the weights of trained models to produce a new set of weights that approximate the weights from retraining. The effect of unlearning is often reflected by the model errors on the deleted data (models cannot predict the deleted data) (Kurmanji et al., 2023; Jang et al., 2022). Another line of work has focused on Differential Privacy (DP) which ensures that user information in training data cannot be inferred (Dwork, 2008; Yu et al., 2021; Li et al., 2021; Anil et al., 2021; Abadi et al., 2016). However, both types of methods require retraining the whole model, which is extremely expensive and time-consuming, especially for large language models and even impacts the task performances (Anil et al., 2021). And thus they can not dynamically tackle sequences of deletion (Jang et al., 2022; Nguyen et al., 2022). To overcome these limitations, we introduce an efficient unlearning method as well as a fusion mechanism to **efficiently** and **dynamically** unlearn sequence of user data.

Our work is also related to model editing (Mitchell et al., 2021; Belinkov et al., 2017; Dai et al., 2021; Wang et al., 2020) while they usually focus on editing the model output based on several given linguistic structures or facts about the world instead of forgetting the required data.

## 3 Efficient Unlearning for LLMs

This section presents our designed **E**fficient **U**nlearning method for **L**LMs (EUL) which could efficiently and dynamically handle a sequence of deletion requests. The overall process is shown in Figure 1. Formally, for a large language model $F(.)$ that is trained on a dataset $D = \{(x, y)\}$ where $x$ is textual data and $y$ is the corresponding label, and a deletion request to forget $D^f = \{(x^f, y^f)\}$, our goal is to learn an updated model $F'(.)$ that satisfies the following (Kurmanji et al., 2023):

$$I(F(D^f); F'(D^f)) = 0$$
$$I(F(D^r); F'(D^r)) = 1 \tag{1}$$

where $D^r = D - D^f = \{(x^r, y^r)\}$ refers to the data we would like to retain, and $I(.)$ is the mutual information. Intuitively, we will update $F(.)$ with $F(.)$ to generate similar output for the data we want to retain while losing all information about making predictions on the data we want to forget.

## 3.1 Learning to Forget via Unlearning Layers

As the scales of current LLMs and the size of training data are usually large, updating all the parameters in the model $F(.)$ (e.g., re-training $F(.)$ on $D_i^r$) becomes extremely expensive. Inspired by recent advances in parameter-efficient fine-tuning (Houlsby et al., 2019; Chien et al., 2023), we model $F'(.)$ by $F(f(.))$ where $f(.; W)$ is an adapter with significant smaller amount of parameters $W$ compared to $F(.)$. And we would only update $f(.)$ to fulfill the unlearning requests.

To effectively achieve the unlearning goals in equation 1, we minimize a selective teacher-student objective where the student model $F'(.) = F(f(.))$ is learned to follow the teacher model $F(.)$ on $D^r$ while disobeyed $F(.)$ on $D^f$:

$$
\begin{aligned}
L_{KL} = &\alpha \sum_{x^r} KL(F(x^r)||F(f(x^r))) \\
&- \sum_{x^f} KL(F(x^f)||F(f(x^f)))
\end{aligned} \tag{2}
$$

where $\alpha$ is a hyper-parameter to balance the trade-off between forgetting $x^f$ and retaining $x^r$. Intuitively, during training, $f(.)$ is leaned to minimize the KL-divergence between the output from the updated model and the original model on the data to retain while maximizing the KL-divergence between the output from them on the data to forget.

To maintain the task performance, we optimize $f(.)$ for the task loss on the retain data:

$$L_{TASK} = \sum_{x^r} l(F(f(x^r)), y^r) \tag{3}$$

where $l(.)$ is the task-related loss, for example, cross-entropy loss, $-\log P(F(f(x^r)))$, for classification tasks.

Furthermore, we also negate the original training objectives used in LLMs (e.g., masked language modeling objective (Raffel et al., 2020)) to forget the knowledge related to the data, in order to forget in pre-trained parameters and ensure that the information in the forgotten data cannot be easily extracted from $F(.)$:

$$L_{LM} = -\sum_{x^f} l(F(f(x^f))) \tag{4}$$

where $l(.)$ is the language model loss used when pre-training $F(.)$, for example, masked language model loss, $-\log P(\hat{x}|x - \hat{x})$ ($\hat{x}$ are the randomly masked tokens). In our experiments, we utilize T5 models (Raffel et al., 2020). Thus we add an extra "*Predict the masked word*" at the beginning of the input for this loss term.

Our final training objective is then the following:

$$L_{EUL} = L_{KL} + \lambda L_{TASK} + \gamma L_{LM} \quad (5)$$

where $\lambda$ and $\gamma$ are hyper-parameters. In practice, following Kurmanji et al. (2023), we alternate the updates for the data to be forgotten and the data to be retained to optimize *min-max* terms in $L_{EUL}$ more stably. Specifically, we iteratively perform an epoch of updates on the data to be retained and then an epoch of updates on the data to be forgotten.

### 3.2   Fusing Unlearning Layers

To dynamically handle a sequence of unlearning requests and derive a unified model that could forget all of the requested data, we then introduce a fusion mechanism that could merge different unlearning layers $f_i(.; W_i)$ which are learned to forget $D_i^f = (X_i^f, Y_i^f)$ in the previous section into a single $f^m(.; W_m)$. Namely, we would like the output of $f^m(.)$ on $D_i^f$ being close to $f_i(.)$:

$$\min_{W_m} \sum_i ||W_m^T X_i^f - W_i^T X_i^f||^2 \quad (6)$$

which is a linear regression problem and has a closed-form solution:

$$W_m = (\sum_i X_i^{f^T} X_i^f)^{-1} \sum_i (X_i^{f^T} X_i^f W_i) \quad (7)$$

Specifically, to derive the weights $W_m$ for the merged unlearning layer $f^m$, we would use the pre-computed inner product matrix of the hidden representations before the unlearning layers in LLMs of the forgotten data $X_i^{f^T} X_i^f$ and then compute $W_m$ following Equation 7.

The fusion mechanism ensures efficiency and privacy as it could be performed without any extra training and only requires storing the inner product matrix of the representations of the data to be forgotten instead of the data itself.

## 4   Experiments

### 4.1   Datasets

We conduct experiments on both classification and generation tasks. For the classification task, we

| Dataset | Task | Train | Dev | Test |
|---------|------|-------|-----|------|
| IMDB | Classification | 20000 | 2000 | 25000 |
| SAMSum | Summarization | 14732 | 818 | 819 |

Table 1: Dataset statistics for IMDB and SUMSum.

utilize the IMDB dataset(Maas et al., 2011), which is a sentiment classification dataset consisting of users' reviews of movies, directors, actors, etc. For the generation task, we use SAMSum (Gliwa et al., 2019), which is a recent popular conversation summarization dataset consisting of conversations between different speakers. The dataset statistics are shown in Table 1.

We choose these two datasets because they are widely used (Wang et al., 2021; Yang et al., 2019; Qin et al., 2023; Ji et al., 2023; Wei et al., 2021; Sanh et al., 2021; Chen et al., 2022) to evaluate large language models and both datasets are related to cases where the user might require to remove their data, for example, removing all the reviews of a specific movie or removing all the conversations from one specific speaker.

In experiments, we use the pre-trained NER models from AllenNLP[2] to extract all the entities (names) in IMDB and directly use the speakers' names in SAMSum and simulate the unlearning requests to remove all the data from or related to certain names. Moreover, we substitute all the names in the dev and test set with special tokens.

### 4.2   Evaluation Metrics

To evaluate the performances, following Kurmanji et al. (2023), we measure several metrics: (1) **Performance on the test set**: The task-related performance on the test set, namely, accuracy for IMDB and ROUGE for SAMSum. This measures whether the unlearning algorithms affect the model performance or not. (2) **Performance on the retained set**: The task-related performance on the data to be retained. This measures whether the unlearning algorithms forget the data that need to be retained. Higher performance means that the model remembers the data that is not to be forgotten. (3) **Performance on the forgot set**: The task-related performance on the data to be forgotten. This measures whether the unlearning algorithms effectively forget the data requested to be forgotten. Lower performance means that the model is better at for-

---

[2] https://demo.allennlp.org/

| Methods | # Forgot Data | Test Set ↑ | Retained Set ↑ | Forgot Set ↓ | MLM Loss ↑ | Time (s) ↓ |
|---|---|---|---|---|---|---|
| *T5-base* | | | | | | |
| Original | - | 93.2 | 100 | 100 | 1.46 | - |
| Re-train | | 92.8 | 100 | 92.5 | 1.52 | 6685 |
| Fine-tune | | **93.0** | 100 | 96.5 | 1.47 | 4200 |
| SISA | 0.5% | 92.4 | 98.2 | 91.5 | 1.54 | 1580 |
| Reverse-Gradient | | 92.0 | 97.3 | 68.6 | 1.56 | 4400 |
| MEND | | 92.2 | 98.5 | 73.5 | 1.60 | **34** |
| EUL† | | **93.0** | **100** | **65.7** | **1.78** | 1200 |
| Re-train | | 92.7 | 100 | 91.6 | 1.55 | 6610 |
| Fine-tune | | 92.8 | 100 | 96.2 | 1.48 | 3950 |
| SISA | 1% | 92.2 | 98.1 | 90.4 | 1.55 | 2930 |
| Reverse-Gradient | | 91.5 | 96.4 | 67.4 | 1.59 | 4166 |
| MEND | | 91.3 | 95.5 | 74.6 | 1.62 | **62** |
| EUL† | | **93.0** | **100** | **64.4** | **1.84** | 1526 |
| Re-train | | 92.1 | **100** | 90.2 | 1.56 | 6026 |
| Fine-tune | | 92.0 | 100 | 95.8 | 1.52 | 3133 |
| SISA | 10% | 91.6 | 98.2 | 88.4 | 1.55 | 2010 |
| Reverse-Gradient | | 91.0 | 96.5 | 65.4 | 1.62 | 3228 |
| MEND | | 90.8 | 94.8 | 76.2 | 1.66 | **328** |
| EUL† | | **92.2** | 99.0 | **57.2** | **2.01** | 1828 |
| *T5-3b* | | | | | | |
| Original | - | 97.0 | 100 | 100 | 1.28 | - |
| Re-train | | 96.6 | 100 | 94.8 | 1.30 | 26855 |
| Fine-tune | | **96.7** | 100 | 96.2 | 1.28 | 20465 |
| SISA | 0.5% | 95.0 | 97.2 | 94.1 | 1.33 | 16503 |
| Reverse-Gradient | | 93.3 | 96.5 | 78.9 | 1.42 | 21826 |
| MEND | | 93.0 | 95.8 | 89.5 | 1.30 | **4980** |
| EUL† | | 96.5 | **100** | **70.2** | **1.66** | 9240 |
| Re-train | | 96.3 | 100 | 94.2 | 1.30 | 25280 |
| Fine-tune | | **96.5** | 100 | 96.0 | 1.28 | 18466 |
| SISA | 1% | 93.8 | 96.8 | 92.7 | 1.35 | 15680 |
| Reverse-Gradient | | 92.5 | 96.0 | 80.1 | 1.46 | 18800 |
| MEND | | 92.8 | 95.0 | 84.4 | 1.48 | **6600** |
| EUL† | | **96.5** | **100** | **67.5** | **1.72** | 9840 |
| Re-train | | 96.0 | 100 | 93.5 | 1.31 | 22140 |
| Fine-tune | | **96.2** | 100 | 94.0 | 1.30 | 16752 |
| SISA | 10% | 93.0 | 95.5 | 92.2 | 1.35 | 14180 |
| Reverse-Gradient | | 91.9 | 95.2 | 68.4 | 1.46 | 17850 |
| MEND | | 92.0 | 94.2 | 78.5 | 1.50 | 12072 |
| EUL† | | 96.0 | **100** | **60.8** | **1.92** | **10460** |

Table 2: Performances on IMDB for T5-base and T5-3B after unlearnling different number of privacy-related data. † refers to our model. All the results are averaged over 5 random runs.

getting the data. (4) **MLM Loss**: The masked language model loses on the data to be forgotten where related entities or actions are masked. This is achieved by adding "*Predict the masked word*" in the beginning. This measure whether the information in the data that needs to be forgotten can be extracted from the LLMs. Higher MLM loss means that it is harder to extract such information from the models. (5) **Updating time**: The time to update the original model in the forgetting process.

## 4.3 Baselines

We compare our EUL with several baseline methods: **Re-train** (Kumar et al., 2022): Re-training the model from scratch on the data to be retained without any forgotten data. **Fine-tune** (Kurmanji et al., 2023): Fine-tuning the original model on the data to be retained without any forgotten data. **SISA** (Kumar et al., 2022): Sharded, Isolated, Sliced, and Aggregated training where multiple models are trained independently on disjoined shards, and its slices and model checkpoints are saved for each

| Methods | # Forgot Data | Test Set ↑ | Retained Set ↑ | Forgot Set ↓ | MLM Loss ↑ | Time (s) ↓ |
|---|---|---|---|---|---|---|
| | | *T5-base* | | | | |
| Original | - | 47.2/23.5/39.6 | 71.4/42.6/62.7 | 70.2/42.2/62.7 | 1.37 | - |
| Re-train | | 46.8/23.0/38.1 | 71.7/42.8/62.4 | 42.4/23.2/42.0 | 1.40 | 28000 |
| Fine-tune | | 46.6/23.2/38.1 | **72.5/44.7/65.2** | 58.8/34.1/54.1 | 1.38 | 27120 |
| SISA | 0.5% | 44.2/22.0/37.4 | 70.5/41.6/60.5 | 41.4/23.0/40.8 | 1.48 | 22582 |
| Reverse-Gradient | | 43.2/20.9/35.8 | 68.8/40.2/58.5 | 42.3/21.4/38.1 | 1.64 | 28800 |
| EUL† | | **46.8/23.0/38.5** | 71.5/42.4/63.3 | **38.4/20.2/37.2** | **1.88** | **17060** |
| Re-train | | 45.4/22.8/37.5 | 72.4/43.0/62.8 | 42.2/22.8/41.6 | 1.44 | 26855 |
| Fine-tune | | **46.4/23.2/38.1** | **72.9/43.6/64.0** | 56.4/31.8/52.7 | 1.40 | 27210 |
| SISA | 1% | 43.1/21.1/36.8 | 69.8/40.2/60.0 | 41.4/23.0/40.8 | 1.50 | 22420 |
| Reverse-Gradient | | 42.0/20.0/34.6 | 68.8/40.2/58.5 | 42.3/21.4/38.1 | 1.64 | 27700 |
| EUL† | | 46.5/22.8/38.0 | 71.5/42.4/63.3 | **35.8/19.0/36.2** | **1.95** | **16820** |
| Re-train | | 44.2/21.2/35.8 | 70.4/41.2/60.5 | 41.4/21.4/40.0 | 1.48 | 26155 |
| Fine-tune | | 45.2/22.1/36.6 | **71.1/42.6/62.9** | 51.5/28.6/50.0 | 1.43 | 27510 |
| SISA | 10% | 41.8/19.6/33.8 | 68.3/38.8/58.8 | 40.2/20.1/38.9 | 1.55 | 20790 |
| Reverse-Gradient | | 40.8/18.4/33.0 | 66.6/38.3/55.5 | 38.0/19.4/36.6 | 1.71 | 27240 |
| EUL† | | **45.8/22.4/37.8** | 70.9/42.0/62.3 | **33.0/18.3/33.0** | **2.23** | **15000** |
| | | *T5-3b* | | | | |
| Original | - | 53.6/29.6/45.1 | 78.5/47.6/66.1 | 74.2/43.5/64.9 | 1.30 | - |
| Re-train | | 52.8/28.8/44.0 | 77.4/46.1/65.4 | 50.4/27.2/43.0 | 1.34 | 84480 |
| Fine-tune | | 53.3/29.0/44.4 | **78.0/47.1/65.8** | 60.2/36.1/55.7 | 1.30 | 83600 |
| SISA | 0.5% | 51.7/27.2/40.8 | 74.8/44.8/63.5 | 49.4/26.8/42.2 | 1.33 | 75000 |
| Reverse-Gradient | | 50.6/25.9/39.9 | 72.8/42.0/62.8 | 44.3/23.1/39.0 | 1.44 | 83200 |
| EUL† | | **53.6/29.4/44.8** | 77.5/46.3/66.6 | **41.0/21.8/38.2** | **1.67** | **60430** |
| Re-train | | 52.0/28.2/42.8 | **76.7/45.8/64.8** | 49.6/26.6/42.1 | 1.35 | 82440 |
| Fine-tune | | 52.5/28.5/43.6 | 76.2/45.5/64.2 | 56.8/32.2/52.4 | 1.32 | 81135 |
| SISA | 1% | 50.0/26.1/38.9 | 72.3/43.1/61.1 | 49.0/25.8/41.1 | 1.38 | 73550 |
| Reverse-Gradient | | 48.6/24.3/37.2 | 70.6/41.5/60.9 | 42.2/22.0/37.7 | 1.45 | 82485 |
| EUL† | | **53.3/29.0/44.4** | 76.4/45.3/64.3 | **38.4/19.9/36.0** | **1.74** | **60880** |
| Re-train | | 50.8/26.4/40.5 | 74.2/45.0/63.2 | 48.2/25.5/41.4 | 1.38 | 81010 |
| Fine-tune | | 51.4/27.2/41.9 | **75.2/45.3/64.0** | 52.1/29.8/49.9 | 1.35 | 81800 |
| SISA | 10% | 48.2/24.5/36.0 | 70.4/40.5/59.6 | 41.2/23.5/40.0 | 1.40 | 70400 |
| Reverse-Gradient | | 44.7/22.0/34.2 | 68.5/40.9/58.8 | 40.9/21.0/36.5 | 1.49 | 82070 |
| EUL† | | **52.0/28.4/42.6** | 74.9/45.0/63.6 | **36.2/18.6/34.7** | **1.78** | **59900** |

Table 3: Performances on SAMSum for T5-base and T5-3B after unlearnling different number of privacy-related data. † refers to our model. All the results are averaged over 3 random runs. The performance on Test, Retained and Forgot Set are ROUGE-1/2/L scores.

slice. When forgetting certain data, the corresponding data point is deleted from its slice, and the model checkpoint up to the data point is used to further retrain the model. **Reverse-Gradient** (Liu et al., 2022): Fine-tuning the original model on both retained data and forgot data while negating the gradient for the forgot data. **MEND** (Mitchell et al., 2021): Editing the model to generate output following the given examples. To adapt the model in the unlearning setting, we reverse the labels for data in classification tasks as input to MEND. However, it is infeasible to apply MEND to summarization tasks as it is hard to design the new output to perform the editing.

## 4.4 Model Settings

For all the experiments, we use T5 models (T5-base and T5-3b) (Raffel et al., 2020) as the backbone models. For SISA, we follow Kumar et al. (2022) to split the dataset. For our unlearning layers, we only tune 0.5% (Chen et al., 2023) of the parameters. The $\alpha = 0.8$, $\lambda = 1.0$ and $\gamma = 0.2$ are selected from grid searching $\{0.1, 0.2, 0.5, 0.8, 1.0\}$. We set the linear decay scheduler with a warmup ratio of 0.06 for training. The maximum sequence length is 128 for IMDB and 800 for SAMSum. The batch size was 256 for base models and 128 for 3b models on IMDB and 8 for base models and 2 for 3b models on SAMSum. The maximum learning

| Methods | Test Set ↑ | Retained Set ↑ | Forgot Set ↓ | Updating Time (s) ↓ |
|---|---|---|---|---|
| Original | 91.8 | 100 | 91.2 | - |
| Re-train | 92.5 | **100** | 12.6 | 6026 |
| Fine-tune | 92.3 | 100 | 26.8 | 3133 |
| SISA | 92.2 | 98.2 | 12.6 | 1510 |
| Reverse-Gradient | 92.8 | 98.6 | 9.0 | 3228 |
| MEND | 92.2 | 97.8 | 16.8 | **328** |
| EUL† | **93.0** | 99.0 | **5.0** | 1828 |

Table 4: Performances on IMDB for T5-base after unlearnling 10% wrong-labeled data. † refers to our model. All the results are averaged over 5 random runs.

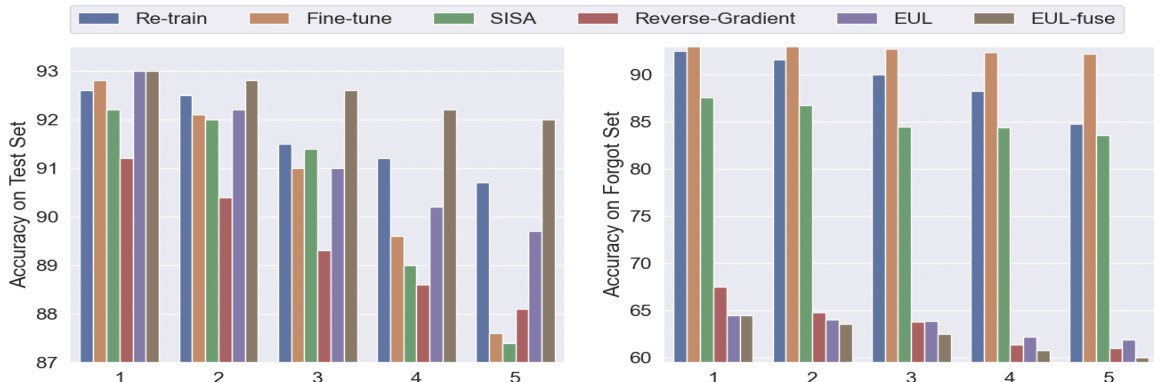

Figure 2: Sequentially unlearnling 1,2,3,4,5 different sets of data for T5-base on IMDB. The results are accuracy on the test set and the accuracy on the forgot set averaging across different orderings. Every single set contains 1% of the training data.

| Metric | EUL | -KL | -TASK | -LM |
|---|---|---|---|---|
| Test Set ↑ | **93.0** | 91.4 | 91.0 | 92.4 |
| Retained Set ↑ | **100** | 100 | 97.4 | 99.0 |
| Forgot Set ↓ | **65.7** | 90.8 | 67.4 | 69.0 |
| MLM Loss ↑ | **1.78** | 1.75 | 1.78 | 1.50 |

Table 5: Performances on IMDB for T5-base after removing 0.5% privacy-related data. We remove one objective at a time from our EUL methods.

| Models | Set 2 | Set 2, 1 | Set 2, 1,3 |
|---|---|---|---|
| Re-train | 92.7/91.4 | 92.5/90.8 | 91.3/90.0 |
| Fine-tune | 92.8/96.0 | 92.1/94.0 | 91.0/93.3 |
| SISA | 92.2/90.4 | 92.0/87.8 | 91.2/85.8 |
| Reverse-Gradient | 91.5/67.9 | 90.5/67.2 | 89.8/66.0 |
| EUL | **93.0**/64.6 | 92.1/64.8 | 91.0/64.2 |
| EUL-fuse | **93.0**/64.6 | **92.8**/62.2 | **92.4**/60.8 |

Table 6: Accuracy on the test/retained set of after unlearning sets of data following a sequence (set 2 -> set 1 -> set 3).

rate was $5e - 5$ and the maximum number of training epochs was set to be 3 or 5. All the experiments were performed using 8 A100 GPUs.

## 4.5   Results

**Unlearning Privacy-related Data on IMDB**   We request the T5-base and T5-3b models that are fine-tuned on the IMDB dataset to unlearn 0.5%, 1% and 10% of the training data. The data to be forgotten is randomly selected based on the names of movies, actors, actresses, directors, etc. For example, the model might need to forget all the data points related to "*Lena Numan*". This simulates the

cases where people/companies request to remove all the data related to them. The performances are displayed in Table 2.

After unlearning the requested data from T5-base models, the re-training method hurts the accuracy (e.g., a 1.1 accuracy drop when forgetting 10% data) on the test set because there is fewer data for training, and the accuracy on the retained set keeps unchanged (100%) probably because the model memorizes the retained data. The accuracy on the forgot set drops after re-training (e.g., 92.5 com-

pared to 100 when unlearning 0.5% of the data), showing that the model is forgetting the requested data, and the masked language model loss increases (e.g., increasing 0.06 when unlearning 0.5% of the data), indicating that it is harder to extract the information of the forgot data after re-training. The fine-tuning method shows better test accuracy with less updating time, however, it is worse in terms of forgetting the data. Even though SISA takes significantly less time (only costing around 1/3 of the time compared to re-training) to derive the updated model that forgets the requested data, it receives lower accuracy on the test and retained set, which means that the model prediction abilities get worse because of failing to remember the retained data. When reversing the gradients for the data to be forgotten, the updated model gets better at forgetting with lower test accuracy. The model editing method, MEND, shows better overall performance on nearly all the metrics but it requires extra data to train a model editing module to edit the original model, making the method hard to be generalized to new models and settings. Our EUL approach boosts all the metrics with faster speed to update the model compared to previous unlearning baselines after removing different numbers of privacy-related data (e.g., achieving the lowest accuracy (65.6%) on forgot set while keeping the best test accuracy (93.0%) and 100% retained accuracy with 1/6 of the updating time compared to re-training when forgetting 0.5% of the data), suggesting that our designed unlearning layers that are learned with tailored objectives could efficiently update the LLMs to forget the required data and remain the abilities to perform the tasks. When the size of the backbone model scales up to 3b, the improvements of our EUL are consistent, indicating that our methods could still forget what the user requests even for larger models that are better at memorizing data.

**Unlearning Privacy-related Data on SAMSum** We unlearn 0.5%, 1% and 10% training data from T5-base and T5-3B models that are fine-tuned on the SAMSum dataset. The data to be forgotten is randomly selected based on the speaker names. For example, the model might need to forget all the conversations from "*Jack*". This simulates the cases where people request to remove all the data generated by them. The performances are shown in Table 3. Similarly, our EUL method consistently achieves the best overall performances by effectively forgetting the requested data while re-

membering the retained data and keeping the test ROUGE scores with significantly less amount of training time. This indicates that our objectives could also be generalized to generation tasks.

**Unlearning Mislabeled Data on IMDB** We also test a setting where the data to be forgotten is those with wrong labels. In experiments, we randomly change the labels for 10% of the training data and then request the model to unlearn their impact. This simulates the cases where we improve the models that are trained on noisy data by unlearning the mislabeled data (Kumar et al., 2022). We report the performances with T5-base models in Table 4. We observe that the accuracy of the test set of the original model is affected by the mislabeled data. And our EUL is the most effective approach to unlearn and remove the negative impact of those mislabeled data to achieve the best test accuracy.

**Sequence of Removals** We test baseline and our methods in a setting where a sequence of unlearn requests are received, i.e., the models need to forget different sets of data sequentially. In experiments, we sequentially unlearn 1,2,3,4,5 sets of data from T5-base model on IMDB dataset. For every unlearn length, we test with all the possible sequences and average the accuracy on the test set and the forgot set. For example, when the length of the forgetting requests are 2 (set 1, set 2), we test on the sequence (set 1 -> set 2) and sequence (set 2 -> set 1) and average the final performances. We show the results (accuracy on the test/retained set) of one possible sequence whose length is 3 (set 2 -> set 1 -> set 3) in Table 6 as an example. Averaged performances over different sequence lengths are visualized in Figure 2. EUL means that we keep one unlearning layer to sequentially unlearn different sets of data and EUL-fuse means that for every set of forgot data we learn separate unlearning layers and then merge them into a single unlearning layer via our proposed fusion mechanism. The results demonstrate that our proposed fusion method that combines different unlearning layers could effectively handle the sequence of deletion (achieving higher accuracy on the test set and lower accuracy on the forgot set.) especially when the sequence length gets longer compared to baseline models.

## 4.6 Ablation Studies

**Removal of Objectives** We perform ablation studies to show the effectiveness of each designed objective in EUL by removing each of them when

| Models | IMDB | SAMSum |
|---|---|---|
| Original | **0.542** | **0.510** |
| Re-train | 0.550 | 0.522 |
| Fine-tune | 0.568 | 0.525 |
| SISA | 0.585 | 0.530 |
| Reverse-Gradient | 0.626 | 0.588 |
| EUL | 0.566 | 0.530 |

Table 7: Accuracy from a trained binary classifier to predict whether an input data belongs to the retained set or the forgot set.

learning the unlearning layers in Table 5. Compared to EUL which utilizes all of the learning objectives, removing each of them would result in a performance drop, which demonstrates every component contributes to the final performance. Specifically, removing $L_{KL}$ would increase the accuracy of the forgot set, indicating that $L_{KL}$ is the main factor to forget the requested data. Removing $L_{TASK}$ from EUL would drop the accuracy on the test set, suggesting that $L_{TASK}$ is essential to maintain task performance. Removing $L_{LM}$ decreases the MLM Loss, showing that $L_{LM}$ is the main objective to avoid the extraction of the requested information.

**Member Inference Attack** We further perform Member Inference Attack (MIA) (Kurmanji et al., 2023) on IMDB and SAMSum when unlearn 1% privacy-related data for T5-base models. Specifically, we test the accuracy of a binary classifier which is trained to predict whether the input data belong to the forgotten set or the retained set based on their representations after the final layer of the T5 model. An accuracy closer to 0.5 means that it is hard for the classifier to predict the groups of the input data. The accuracies are shown in Table 7. We found that the classifiers could not converge so well on the training set and always had a low accuracy on the test set both before and after unlearning (e.g., 0.542 before unlearning and 0.566 after our EUL unlearning on IMDB). These showed that the randomly deleted data could not be easily inferred both before and after our EUL unlearning.

## 5 Conclusion

In this work, we propose EUL, an efficient unlearning method for LLMs that could efficiently and effectively unlearn the user-requested data via learning unlearning layers through the selective teacher-student objective. We further introduce a fusion mechanism that could merge different unlearning layers into one unified layer to dynamically unlearn a sequence of data. Experiments on different settings (different datasets, different model sizes, different forget set sizes) demonstrated the effectiveness of our proposed EUL method compared to state-of-the-art baselines.

## 6 Limitations

In this work, we mainly perform experiments on T5-base/3b models with fine-tuned tasks. We encourage future work to explore how to update different backbone models with larger sizes such as LLAMA models or even close-sourced models like ChatGPT to forget the requested data such asemn privacy-related data, toxic data, or misinformation in the pre-training corpus. Also, we mainly follow the previous work to measure the unlearning through performance on the test set, retained set, and forgot set, together with the MLM loss. Future work might explore how to evaluate unlearning methods more comprehensively, such as whether the model could recall forgotten content or whether methods would make forgotten data identifiable. In addition, we perform all the experiments in simulated settings. Future work might apply our methods to real-world applications to deal with actual use cases or introduce new benchmarks for evaluating unlearning methods.

## Acknowledgment

We would like to thank all reviewers and the SALT Lab for their valuable feedback. This work was partially sponsored by NSF grant IIS-2247357 and IIS-2308994.

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
