# OpenReview forum: "Unlearn What You Want to Forget: Efficient Unlearning for LLMs"
_EMNLP/2023/Conference — EMNLP 2023 Main_

### Official Review · Reviewer_41eo · 2023-08-01

**Soundness:** 3

**Excitement:**

3: Ambivalent: It has merits (e.g., it reports state-of-the-art results, the idea is nice), but there are key weaknesses (e.g., it describes incremental work), and it can significantly benefit from another round of revision. However, I won't object to accepting it if my co-reviewers champion it.

**Paper Topic And Main Contributions:**

This paper mainly focuses on handling unlearning requests for LLMs. An efficient unlearn method, that freezes the LLMs' parameters while inserting trainable unlearning layers into transformer layers to forget requested data,  is proposed combined with a fusion mechanism for fusing the multiple unlearning layers trained by obtained by a sequence of unlearning requests. The experiments show the obvious improvement achieved by the proposed method.

**Questions For The Authors:**

1. Can you show the time series experimental results of the sequence of forgotten requests?
2. With the rapid of the LLMs' size, can unlearning layers fit well during training and remain the performances?

**Reasons To Accept:**

1. Due to the rapid development of large-scale language models, this topic needs to be focused on.
2. This paper proposed an efficient unlearning method for LLMs to forget data while retaining the performances of the model.
3. The experiments show the obvious improvement achieved by the EUL.

**Reasons To Reject:**

1. The effective of fusion mechanism has not been evaluated in the experiments.
2. For the sequence of forgotten requests, the results of experiments during the temporal process should be presented, the same as the continual learning, in order to verify that the model will not recall previously forgotten content.
3. The proposed EUL is validated only on specific LLMs (T5-base and T5 3b), and more comparisons with different LLMs in different sizes and backbones need to be conducted. Especially, with the rapid of the LLMs' size, can unlearning layers fit well during training and remain the performances?
4. Technical writing skill needs to be improved.

**Reproducibility:**

3: Could reproduce the results with some difficulty. The settings of parameters are underspecified or subjectively determined; the training/evaluation data are not widely available.

**Reviewer Confidence:**

2: Willing to defend my evaluation, but it is fairly likely that I missed some details, didn't understand some central points, or can't be sure about the novelty of the work.

---

> ### Author Rebuttal · Authors · 2023-08-28
>
> Thanks for your reviews. We hope our response below could solve your concerns.
>
> ### On the evaluation on the fusion mechanism
> The fusion mechanism is proposed to handle the sequence of unlearning operations. We evaluate the effectiveness of our proposed fusion mechanism through the experiments where we gradually forget a set of data in an sequential way (sequence of removals in Section 4.5) and the results are shown in Figure 2. The results demonstrate that our proposed fusion method could effectively handle the sequence of deletion (achieving higher accuracy on the test set and lower accuracy on the forgot set compared to baseline models) especially when the sequence length gets longer compared to baseline models.
>
> ### The results for sequence of removals
>
> For every length of the removals, we test all the possible sequences and report the average performance. For example, when removing 2 sets (set 1, set 2), we test the performances of different removing sequences: set 1->set 2 and set 2->set 1. And this kind of evaluation is used in previous continual learning work [1]. We would add all the performances of every sequence in the Appendix in the revised version.
>
> [1] Huang, Yufan, et al. "Continual Learning for Text Classification with Information Disentanglement Based Regularization." Proceedings of the 2021 Conference of the North American Chapter of the Association for Computational Linguistics: Human Language Technologies. 2021.
>
> ### On the concern and question about effectiveness on different backbone and larger models
>
> We mainly performed experiments on T5-base/3b to illustrate the effectiveness of our introduced unlearning methods. We did not test on larger LLMs because of the computation limit. Actually, the designs of our ELU architectures and methods are agnostic to different types of pre-trained models. In our early experiments, we have tested on RoBERTa-base models on the classification tasks (IMDB) and we observed the gains of our methods (after unlearning 10\% of the IMDB data, the accuracy on the test set, retained set and forgot set are 0.946, 1.0, 0.428).
>
> What’s more, from our experiments on models with different sizes, we could observe a consistent forgetting ratio (e.g., 36\% on T5-base and 34.9\% on T5-3b on classification tasks when forgetting 10\% of the data.).
>
> Note that, as our model architectures are similar to adapters [1] which have been shown to be applicable to most recent larger models like LLAMA models, we think our EUL would fit well for larger models.
>
> [1] Zhang, Renrui, et al. "Llama-adapter: Efficient fine-tuning of language models with zero-init attention." arXiv preprint arXiv:2303.16199 (2023).
>
> ### On the questions about the time series results of the sequence of forgetting requests
> In our experiments, we have actually tested on all the possible sequences for a given sequence length. For example, when the length of the forgetting requests are 2 (set 1, set 2), we have tested on the sequence (set 1->set 2) and sequence (set 2->set 1) and averaged the final performances. We would add all the results on different sequences in the revised version.
>
> Here, we show the results (accuracy on the test/retained set) of one possible sequence whose length is 3 (set 2 -> set 1 -> set 3) below as an example.
> | Methods          | set 2     | set 2,1   | set 2,1,3 |
> |------------------|-----------|-----------|-----------|
> | Re-train         | 92.7/91.4 | 92.5/90.8 | 91.3/90.0 |
> | Fine-tune        | 92.8/96.0 | 92.1/94.0 | 91.0/93.3 |
> | SISA             | 92.2/90.4 | 92.0/87.8 | 91.2/85.8 |
> | Reverse-Gradient | 91.5/67.9 | 90.5/67.2 | 89.8/66.0 |
> | EUL              | 93.0/64.6 | 92.1/64.8 | 91.0/64.2 |
> | EUL-fuse         | 93.0/64.6 | 92.8/62.2 | 92.4/60.8 |

---

### Official Review · Reviewer_sR7i · 2023-08-02

**Typos Grammar Style And Presentation Improvements:** 1. [Line 002] on and => remove "on"
2…
**Soundness:** 3

**Excitement:**

3: Ambivalent: It has merits (e.g., it reports state-of-the-art results, the idea is nice), but there are key weaknesses (e.g., it describes incremental work), and it can significantly benefit from another round of revision. However, I won't object to accepting it if my co-reviewers champion it.

**Paper Topic And Main Contributions:**

This paper proposes to unlearn certain data by adding unlearning layers to the original model. During training, only the unlearning layers are tuned to minimize the final objective which contains three parts: L_KL, L_TASK, and L_LM. Moreover, the paper also studies fusing different unlearning layers to handle multiple forgetting requests. The method is tested on one classification task and one generation task.

**Questions For The Authors:**

1. [Section 3.1] For the language modeling part, what is the percentage of masked tokens? Are they randomly masked?
2. [section 4.4] What is the architecture of the unlearning layers?
3. [Section 4.5] According to Table 1, EUL has a much lower accuracy than the re-train model on the forgot set (e.g. 65.7 vs 92.5 for T5-base). From my understanding, an ideal unlearning model should mimic the re-train model's behavior and make it hard to tell the difference between the forgotten data and unseen data. Will the extremely low accuracy make the forgotten data easily detectable?
4. [Section 4.6] In Table 5, why does removing LM have such a large impact on the retained set? The accuracy drops from 100 to 69.

**Reasons To Accept:**

1. Machine unlearning is a very interesting and underexplored area. The proposed method is promising when removing mislabeled data is needed.
2. The proposed method is straightforward and easy to implement. Also, the method can deal with a sequence of forgetting data requests.

**Reasons To Reject:**

I have three main concerns:
1. I doubt the proposed method can work as well as they claim in the paper under the privacy protection scenario. The proposed method encourages getting a high error on the forgotten data. However, it also makes the forgotten data identifiable and vulnerable to attacks. I think adding metrics like Membership Inference Attack (MIA) is necessary.
2. The metrics used in the paper are not sufficient and some of them are inappropriate. As I mentioned in 1, the paper lacks metrics like MIA to show the model's performance on unlearning privacy-related data. Moreover, I do not understand why the MLM loss on the forgotten data is used as both an optimization objective and an evaluation metric.
3. The paper only reports results on two datasets. I hope to see the results on some text completion tasks to check if the model can really avoid generating user-related private information.

**Reproducibility:**

3: Could reproduce the results with some difficulty. The settings of parameters are underspecified or subjectively determined; the training/evaluation data are not widely available.

**Reviewer Confidence:**

3: Pretty sure, but there's a chance I missed something. Although I have a good feel for this area in general, I did not carefully check the paper's details, e.g., the math, experimental design, or novelty.

---

> ### Author Rebuttal · Authors · 2023-08-28
>
> Thanks for your insightful comments. We humbly think that some concerns might be caused by misunderstanding, which we will explain in detail below. We hope that our response can clarify the misunderstandings and you can consider our work more favorably in the ratings.
>
> ### On the Membership Inference Attack (MIA)
> We actually have evaluated the MIA attack in our preliminary studies when performing unlearning on the IMDB dataset where we tried if we could learn a binary classifier to predict whether a data belongs to the forgotten set or retained set based on the representations. And we found that the classifiers could not converge so well on the training set and always had a low accuracy on the test set both before and after unlearning (e.g., 0.542 before unlearning and 0.566 after unlearning). The results are shown in the table below. These showed that the randomly deleted data could not be easily inferred both before and after the unlearning. As a result, we did not present the studies in the main paper. We would add this study and results to the revised version.
> | Models | Original Model | Re-train | Fine-tune | SISA  | Reverse-Gradient | EUL   |
> |--------|----------------|----------|-----------|-------|------------------|-------|
> | MIA    | 0.542          | 0.550    | 0.568     | 0.585 | 0.626            | 0.566 |
>
> ### On the evaluation metrics
> As stated in the limitation section in the paper, we mainly follow the previous work to measure the unlearning through performances on the test set, retained set, and forgot set, together with the MLM loss. While there does not exist  a well-defined evaluation metric, we hope our study could facilitate future research on evaluating unlearning LLMs. The reason why we are utilizing MLM Loss as an additional metric is that we want to measure whether the information in the data that needs to be forgotten can be extracted from the LLMs through the token predictions. This factor has not been considered before while being raised up as a risk issue for LLMs where identifiable information like names, phone numbers, email addresses, and even bank account numbers can be easily extracted from LLMs through token predictions in a given context [1]. And we optimize such objectives during training. This is similar to MSE Loss when training regression models where L2 distance is used in both optimization and evaluation.
> [1] Carlini, Nicholas, et al. "Extracting training data from large language models." 30th USENIX Security Symposium (USENIX Security 21). 2021.
>
> ### On the concern about the selection of tasks
> We utilize two representative tasks (classification and summarization) to evaluate the effectiveness of the unlearning methods. And we utilize the MLM Loss to measure whether the deleted personal information (such as names, address) could still be inferred in a given masked sequence. Our unlearning methods could achieve a higher MLM loss which means that it is harder to extract deleted information from the models. For example, after unlearning an utterance (“John: I will pick Bob up on Main Street.”)  In the conversation summarization dataset, the model after our EUL could not predict the “Bob” for the masked sequence (“John: I will pick [MASK] up on Main Street.”) which prevents the personal information from being extracted from LLMs. Again, as a starting point, we perform all the experiments in simulated settings. However, our methods could be easily applied to real-world applications to deal with actual use cases.
>
> ### On the question about the masked tokens during the training
> The tokens are randomly selected to be masked and the ration is 15\%.
>
> ### On the question about the  architecture of the unlearning layers
> We utilize the two feed-forward linear layers in one unlearning layer.
>
> ### On the question about detecting forgot data
> As stated in the response to the first concern, we have performed the MIA attack in our early experiments and we found that they can not be easily inferred both before and after the unlearning. We would add the studies in our revised version.
>
> ### On the performance on the retained set after removing the LM loss
> We made a typo for the number and the performance on the retained set should be 99.0. We would correct the typo in the revised version.

---

### Official Review · Reviewer_wd9K · 2023-08-02

**Soundness:** 4

**Excitement:**

4: Strong: This paper deepens the understanding of some phenomenon or lowers the barriers to an existing research direction.

**Paper Topic And Main Contributions:**

This paper addresses the problem of effectively forgetting some information in large language models, in order to prevent some privacy leak or bias. The authors propose an unlearning layer to be trained without re-training on the entire model. Experiments are performed on both generative and discriminative tasks to show the effectiveness of the model.

**Questions For The Authors:**

In Table5, it is shown that different losses contribute to different aspect of the performance. Does this finding still hold in the summarization dataset?


**Reasons To Accept:**

1. After trained on huge amount of data, deleting certain information that might be hurtful, toxic, or sensitive without re-training the model is very important. The authors propose a simple but effective method to teach the model unlearn without any retrain steps would be very beneficial for future LLM development.
2.The fused learning mechanism allows the model to remove the unneeded information in a sequence. Figure 2 shows the effectiveness of this component compared with the original version.
3.The authors conduct several experiments to test the model on both classification and generation tasks under two base models. Experiment results show that the EUL method helps to save the time and assist the model unlearn certain information.
4. The motivation and method parts are clear and the experiments are reasonable.


**Reasons To Reject:**

1.The main concern of this paper is that they did not perform experiments on LLMs such as LLAMA but only adopt T5 as the base model. However, as shown in Table2 and Table3, it seems the performance on the forget set increases when the base model grows larger (T5-3b v.s. T5-base). This result shows that, as the language model grows larger, it gets harder for them to forget. This is reasonable because the larger the language model is, the more the model knows. Therefore, it would provide the essential insight if the proposed method would still be beneficial in the recent proposed LLMs, and how much will it forget under this method.

2. Even though the model has performance drop on the forgot data, it’s still very hard to say whether it truly ‘forgets’ the data, considering there always remain some common knowledge overlap between forgot data and the training portion. As in the limitation part states, the experiments are performed in a simulated setting, it would be more convincing if the users can provide some real cases to analyse which data is easily to be forgotten and which is not.


**Reproducibility:**

4: Could mostly reproduce the results, but there may be some variation because of sample variance or minor variations in their interpretation of the protocol or method.

**Reviewer Confidence:**

4: Quite sure. I tried to check the important points carefully. It's unlikely, though conceivable, that I missed something that should affect my ratings.

---

> ### Author Rebuttal · Authors · 2023-08-28
>
> Thanks for your positive assessment and constructive feedback.
>
> ## On the concern about effectiveness for larger LLMs
> We performed experiments on T5-base and T5-3b as working examples to illustrate the effectiveness of our introduced unlearning methods. We did not test on recent larger LLMs because of the computation limit. However, from our experiments on smaller (base) and larger (3b) models, we could observe consistent forgetting ratio (e.g., 36\% on T5-base and 34.9\% on T5-3b on classification tasks when forgetting 10\% of the data.). As a result, we think our method would still work for recent proposed LLMs.
>
> ## On the concern about evaluations about the forgetting
> Our work is a starting point on the important issues for deploying large language models about how to unlearn the knowledge/data in large LLMs. While there are no well-defined test sets and evaluation metrics, we evaluate our methods with metrics used in all of the previous work about unlearning LLMs on both classic classification and generation datasets which contain privacy information to simulate the user cases.  In terms of the metrics, besides the  performance on different sets used in previous studies, we include MLM loss to further reflect whether generative LLMs could still predict user information that needs to be forgotten. And our results demonstrate that our methods could forget the knowledge effectively and efficiently even in a sequential setting which previous methods fail to handle.
> What’s more, in our early studies, we also evaluated the MIA attack when performing unlearning on the IMDB dataset where we tried if we could learn a binary classifier to predict whether a data belongs to the forgotten set or retained set based on the representations. And we found that the classifiers could not converge so well on the training set and always had a low accuracy on the test set both before and after unlearning (e.g., 0.542 before unlearning and 0.566 after unlearning). The results are shown in the table below. These show that the randomly deleted data could not be easily inferred both before and after the unlearning. As a result, we did not present the studies in the main paper. We would add this study and results to the revised version.
>
> | Models | Original Model | Re-train | Fine-tune | SISA  | Reverse-Gradient | EUL   |
> |---------|--------------|----------|-----------|-------|------------------|-------|
> | MIA    | 0.542          | 0.550    | 0.568     | 0.585 | 0.626            | 0.566 |
>
> We hope our preliminary study could raise awareness of the community on this important problem about forgetting the required information in LLMs and facilitate future research on unlearning LLMs.
>
> ## On the question about the ablation studies on summarization dataset.
> Yes, the findings still hold for summarization dataset. We would add the additional results in the revised version.

---

### Meta-Review · Area_Chair_zpDQ · 2023-09-19

**Recommendation:** 4

**Metareview:**

This paper addresses the problem of effectively forgetting some information in large language models

The authors propose a simple but effective method to teach the model unlearn without any retrain steps.
Several experiments are conducted to test the model on both classification and generation tasks under two base models.
The motivation and method parts are clear and the experiments are reasonable.
Machine unlearning is a an interesting and underexplored area. The proposed method is promising when removing mislabeled data is needed.

Experiments should have been performed on larger LLMs such as Llama.
The contribution of individual components could be evaluated better.
Several important questions are not considered - whether the model could recall previously forgotten content, and whether the method makes forgotten data identifiable.

---

### Decision · Program_Chairs · 2023-10-07

**Decision:**

Accept-Main

**Comment:**

This paper addresses the problem of effectively forgetting some information in large language models

The authors propose a simple but effective method to teach the model unlearn without any retrain steps.
Several experiments are conducted to test the model on both classification and generation tasks under two base models.
The motivation and method parts are clear and the experiments are reasonable.
Machine unlearning is a an interesting and underexplored area. The proposed method is promising when removing mislabeled data is needed.

Experiments should have been performed on larger LLMs such as Llama.
The contribution of individual components could be evaluated better.
Several important questions are not considered - whether the model could recall previously forgotten content, and whether the method makes forgotten data identifiable.